# Relationship of vitamin D, fibrinogen and their ratio with acute coronary syndrome: A comparative analysis of unstable angina, NSTEMI, and STEMI

Himayat Ullah[1], Sarwat Huma[2], Muhammad Ashraf[1], Nafisa Tahir[3], Ghulam Yasin[1], Mohammed Yunus[4*], Hossam Shabana[1,5], Mohamed Ahmed Muharram[5], Abdulrahman H. Shalaby[5], Ahmed Ali Hassan Ali[5], Mohamed Elwan Mohamed Mahmoud[6], Ahmed Mahrous Ahmed Ibrahim[5], Ahmed Farag Abd Elkader Elbwab[5], Ahmed Mohamed Ewis Alhawy[5], Ahmed Ahmed Mohamed Abotaha[5], Mahmoud Ezzat Abdelraouf[5], Mohammed S. Imam[5], Hossam Aladl Aladl Aladl[5], Taiseer Ahmed Shawky[5], Ashraf Mohammed Said[5], Mahmoud Saeed Mahmoud[6], Kazem Mohamed Tayee[5], Reda Fakhry Mohamed[5], Ali Hosni Farahat[5], Mohammad Mossaad Abd Allah Alsayyad[6], Hesham El Sayed Lashin[5], Essam Yehia Ali Aggour[5], Hazem Sayed Ahmed Ayoub[5], Ayman Mohamed Salem Ahmed[5], Marwan Sayed Mohamed Ahmed[5], Abdelrahman E. Metwally[7], Ahmed Saeed Abdelaziz[7], Tamer Ahmed Fouad Mohammed Hassan[7]

**1** College of Medicine at Shaqra, Shaqra University, Saudi Arabia, **2** Health Services Academy, Hayatabad Medical Complex, Peshawar, Islamabad, Pakistan, **3** NUST School of Health Science, National University of Science and Technology, Islamabad, Pakistan, **4** Kabul University of Medical Sciences (KUMS), Kabul, Afghanistan, **5** Faculty of Medicine, Al-Azhar University, Cairo, Egypt, **6** Faculty of Medicine, Al-Azhar University, New Damietta, Egypt, **7** Cardiovascular Department, Faculty of Medicine, Al-Azhar University, Cairo, Egypt

* m_yunus6061@yahoo.com

## Abstract

### Background

There is emerging evidence suggesting that vitamin D and fibrinogen play contrasting roles in ACS pathophysiology and their combined impact, expressed as the vitamin D/fibrinogen ratio, can be a potential biomarker for ACS severity.

This study aimed to investigate the relationship between vitamin D, fibrinogen, and their ratio with ACS types, and assess their potential as risk stratification biomarkers.

### Methods

This multicenter observational study was conducted in tertiary care hospitals in Afghanistan, Egypt, and Pakistan, including 300 ACS patients. Serum vitamin D and fibrinogen levels were measured using electrochemiluminescence immunoassay and the Clauss method, respectively. Statistical analyses included ANOVA, Kruskal-Wallis, post-hoc Games-Howell tests, Spearman's correlation, Fisher's Z-test, and multivariable logistic regression.

**Data availability statement:** All relevant data are within the manuscript and its Supporting Information files.

**Funding:** The author(s) received no specific funding for this work.

**Competing interests:** The authors have declared that no competing interests exist.

## Results

Vitamin D levels were significantly lower ($p < 0.001$) and fibrinogen levels significantly higher ($p < 0.001$) in STEMI patients compared to NSTEMI and UA. The vitamin D/fibrinogen ratio showed a stronger correlation with ACS severity (Spearman's rho $= -0.45$, $p = 0.01$) than vitamin D alone ($-0.41$, $p = 0.01$), but this difference was not statistically significant (Fisher $Z = 0.34$, $p = 0.73$). Logistic regression revealed that a 1 nmol/L increase in vitamin D reduced ACS severity by 7.1% ($p = 0.043$), while a unit increase in the vitamin D/fibrinogen ratio reduced severity by 6.2% ($p = 0.048$).

## Conclusion

The contrasting effects of vitamin D and fibrinogen can prove useful biomarkers and modifiable risk factors for ACS. The superiority of the vitamin D/fibrinogen ratio over vitamin D only, however, needs further validation in larger studies.

## Introduction

Acute Coronary Syndrome (ACS) is a clinical picture caused by a sudden reduction in cardiac blood flow leading to an acute myocardial ischemia/infarction. It is usually characterized by a set of typical symptoms including typical chest pain radiating to the neck, jaw and/or arm, nausea and diaphoresis. Based on the ECG findings and cardiac enzymes, ACS is further classified into unstable angina (UA), non-ST segment elevation myocardial infarction (NSTEMI) and ST-segment elevation myocardial infarction (STEMI). For simplification and clarity we can state that, STEMI is the most severe form among these, as it results from the complete blockade of the corresponding coronary artery (causing transmural necrosis), followed by NSTEMI which is caused by incomplete blockade with elevated cardiac enzymes (CK-MB and Troponins) while UA is the mildest one which is caused by incomplete blockade without rise in cardiac enzymes [1,2]. Table 1 below differentiates the different ACS types. ACS remains one of the most prevalent global health issues causing significant morbidity and mortality despite continuous advancement in the management and reperfusion therapies [3]. This fact warrants the necessity of disease prevention and risk control along with the active management of the disease. Several conventional risk factors have widely been studied in medical research including smoking, obesity and dyslipidemias, diabetes and hypertension and so on. Apart from these, there is growing evidence that certain novel biomarkers are also shown to play a critical role in the development of coronary artery disease and coronary syndromes. Two of these novel biomarkers include vitamin D and serum fibrinogen, which have shown a contrasting effect on the development of coronary artery disease [4–6].

Vitamin D, classically known as a calcium-regulatory vitamin responsible for bone health, has drawn attention with its protective effect on the endothelium and anti-inflammatory and anti-thrombotic properties in recent studies [7–9]. Low vitamin D levels contribute to endothelial dysfunction, inflammation and pro-thrombotic states

**Table 1. Summary of vascular occlusion in acute coronary syndrome (ACS) types.**

| ACS Type | Vessel occlusion and clot type | Clinical Significance |
|---|---|---|
| Unstable Angina (UA) | Partial, non-occlusive, platelet clot | No myocardial necrosis, responds well to antiplatelet therapy |
| NSTEMI | Partial but significant occlusion, with platelet and fibrin clot | Myocardial injury requires aggressive antithrombotic therapy |
| STEMI | Complete occlusion with platelet-poor but fibrin and RBC-rich dense clot | High-risk, urgent reperfusion needed |

by impaired nitric oxide synthesis, interleukin 6, tumor necrosis factor-α and higher tissue factor expression. This effect signifies the importance of vitamin D in coronary health, and its deficiency may pose a significant risk towards atherosclerotic coronary artery diseases and their complications [10]. Although the above-cited studies favour the protective role of vitamin D on vascular and cardiac health, its precise role in ACS needs further investigation and workup. For reference, vitamin D levels below 50 nmol/L are referred to as deficiency and levels below 30 nmol/L as severe deficiency [11].

Fibrinogen, a glycoprotein and a member of the clotting factors family (Factor 1), has been studied as one of the non-conventional risk factors and a biomarker of coronary artery disease (CAD). Studies have shown fibrinogen as an inflammatory marker, prothrombotic and inhibitor of fibrinolysis [12,13]. Furthermore, fibrinogen has been shown to promote endothelial dysfunction and vascular inflammation, further exacerbating the risk of cardiovascular events [14]. All of these effects might have a critical role in coronary atherogenesis and thrombosis and thus relate to the severity of acute coronary syndrome (ACS).

Although the effects of vitamin D and fibrinogen on vascular health and coronary arteries have been studied individually, their combined effect has never been studied previously [15]. These contrasting effects of vitamin D and serum fibrinogen may give rise to the hypothesis that the interplay of these two important biomarkers could affect the course and severity of CAD. Vitamin D deficiency and high serum fibrinogen levels may play a synergistic role in coronary vascular inflammation, endothelial dysfunction, platelet aggregation and atherosclerosis, thus adversely affecting the ACS severity.

The rationale behind this study is the growing evidence supporting the contrasting effects of vitamin D and serum fibrinogen on CAD and ACS when studied individually and independently. This complementary effect of vitamin D and serum fibrinogen gives rise to the idea of assessing their combined effect on ACS subtypes in the form of a ratio reflecting the net thrombo-inflammatory status of ACS patients.

The primary objective of this study is to investigate the relationship between vitamin D, fibrinogen, and their ratio with ACS types (Unstable angina, NSTEMI, and STEMI). Secondarily, we also aim to compare the strengths of the association between ACS and vitamin D and serum fibrinogen individually, as well as their ratio. Thirdly, we aim to assess the potential utility of the vitamin D/fibrinogen ratio as a biomarker of risk stratification in ACS.

## Materials and methods

This observational study was conducted in three tertiary care hospitals each belonging to three different countries, Pakistan, Egypt, and Afghanistan in accordance with the ethical principles of the Declaration of Helsinki, from 20th April 2022–15th August 2024. Before the commencement of the study, ethical approval was obtained from the Institutional Review Board (reference number 599/HEC/B&PSE/2021, dated 15th Feb, 2022). Sample size was calculated through an internationally recognized, standardized sample size calculator, based on ACS prevalence in the general population (4%), with a minimum required sample of 60 patients. A non-probability convenience sample of 100 patients (a total of 300) from each country, presented with ACS was recruited for the study. Using G*Power version 3.1, a one-way ANOVA with 3 groups (Unstable Angina, NSTEMI, STEMI), $\alpha = 0.05$, and total sample size of 300 yields a statistical power of 98% to detect a medium effect size (Cohen's $f = 0.25$), confirming that the study is adequately powered to identify meaningful differences across groups.

The inclusion criteria were all the patients above 18 years with confirmed ACS. Patients with severe liver or kidney disease, malignancies and patients taking anticoagulants and/or vitamin D supplementation were excluded from the study. A written informed consent was taken from each patient explaining the study process, including the potential benefits to participants and the community, possible risks, and measures to ensure confidentiality and data anonymization.

All the patients were subjected to detailed history, examination and followed by investigations. ACS was confirmed using ECG, cardiac enzymes, echocardiography and angiography wherever required. Serum vitamin D (25-hydroxyvitamin D) levels were measured using electrochemiluminescence immunoassay (ECLIA) and Fibrinogen levels by the Clauss method. The blood samples were collected at the time of admission, within the first three hours of admission to the facility.

## Statistical analysis

The Statistical Package for Social Sciences version 27 (IBM Corporation; Armonk, NY, USA) and MS Excel (Microsoft Corporation, Washington, USA) were used to analyse the data. Mean±SD were used for continuous variables like vitamin D and fibrinogen while frequencies (%) were for categorical variables like gender and type of ACS. Analysis of variance (ANOVA) followed by the Kruskal-Wallis test was performed to show the difference among the ACS groups. The Bonferroni post hoc (for homogenous variances) and Games-Howell post hoc (for unequal variances) as tested by Levene's test, were performed to evaluate the inter-group pairwise differences in vitamin D, fibrinogen levels, and the vitD/fibrinogen ratio (Unstable angina VS NSTEMI VS STEMI). This approach ensured rigorous control over Type I error in both parametric and non-parametric contexts. Spearman's rank correlation was employed to assess the relationship between the vitamin D/fibrinogen ratio and ACS severity. Multivariable logistic regression was performed to adjust for potential confounders such as age, gender, diabetes, hypertension, and lipid profiles. A p-value <0.05 was considered statistically significant.

## Results

This analytical study included 300 ACS patients from three different countries, namely Afghanistan, Egypt and Pakistan, each contributing 100 patients to the cohort. The basic demographic and biochemical characteristics of the patients are summarized in Table 2.

**Table 2. Baseline characteristics of study participants.**

| Variable | Unstable Angina (n=45) | NSTEMI (n=163) | STEMI (n=92) | Total |
|---|---|---|---|---|
| Age (Mean±S.D) | 58.33±12.04 | 58.11±12.11 | 55.04±15.31 | 57.2±13.21 |
| **Gender N (%)** | | | | |
| Male | 23 (51%) | 82 (50.3%) | 64 (69.6%) | 169 (56.3%) |
| Female | 22 (49%) | 81 (49.7%) | 28 (30.4%) | 131 (43.7%) |
| **Risk Factor N (%)** | | | | |
| Smoking | 22 (48.9%) | 51 (31.3%) | 49 (53.3%) | 122 (40.7%) |
| Dyslipidemia | 19 (42.2%) | 86 (52.8%) | 38 (41.3%) | 143 (47.7%) |
| Diabetes Mellitus (%) | 10 (22.2%) | 59 (36.2%) | 27 (29.3%) | 96 (32%) |
| Hypertension (%) | 9 (20%) | 29 (17.8%) | 23 (25%) | 58 (19.3%) |
| Vitamin D (nmol/L) | 65.36±22.53 | 43.24±15.44 | 35.59±19.84 | 44.21±20.3 |
| Fibrinogen (g/L) | 3.52±1.04 | 4.71±1.39 | 4.80±1.37 | 4.56±1.37 |
| Vitamin D/fibrinogen ratio | 19.71±8.32 | 10.15±5.15 | 7.97±4.85 | 10.91±6.80 |
| **Nationality N (%)** | | | | |
| Afghanistan | 17 (37.8%) | 62 (38%) | 21 (22.9%) | 100 (33.3%) |
| Egypt | 15 (33.3%) | 50 (30.7%) | 35 (38%) | 100 (33.3%) |
| Pakistan | 13 (28.9%) | 51 (31.3%) | 36 (39.1%) | 100 (33.3%) |

Regarding the vitamin D status of the cohort, 31% of unstable angina patients, 67.5% of NSTEMI patients, and 79% of STEMI patients were vitamin D deficient (< 50nmol/L).

Analysis of Variance (ANOVA) followed by the Kruskal-Wallis test showed a significant difference among the three severity groups regarding vitamin D levels, fibrinogen levels, and their ratios. Post hoc pairwise comparisons with Bonferroni and Games-Howell adjustments confirmed statistically significant differences in the vitamin D/fibrinogen ratio between all ACS subtypes (UA, NSTEMI, STEMI; all p < 0.01). The ratio showed a stepwise decline from UA to STEMI, supporting its association with ACS severity. Similarly, serum vitamin D levels were significantly lower in STEMI compared to NSTEMI and UA (p < 0.001). Fibrinogen levels were significantly higher in NSTEMI and STEMI compared to UA (p < 0.001), though the difference between NSTEMI and STEMI was not statistically significant (p = 1.000 (Bonferroni) and 0.89 (Games-Howell)). Brown-Forsythe test showed no significant difference among the patients from the three countries on the basis of vitamin D, fibrinogen and their ratio (P > .05 for all the three). To further mitigate the risk of selection bias, stratified sensitivity analyses across the three study sites were also performed, which showed consistent trends in biomarker levels and their associations with ACS severity across the three countries. These statistics are summarized in Table 3 below.

The strength of the correlation (Spearman Rho) showed a significant correlation between vitamin D, fibrinogen, and their ratio, and ACS types (P < .01). Pairwise comparison of the strength of each correlation (Fisher Z test) showed a significantly stronger correlation of vitamin D and vitamin D/fibrinogen ratio with ACS types as compared to fibrinogen (P < .05). However, although the correlation vitamin D/fibrinogen ratio to ACS types was stronger (0.45) as compared to that of vitamin D alone (0.41), it was not significantly stronger statistically (Fisher Z value 0.34, P value 0.73). These statistics are summarized in Table 4.

Boxplot showing the distribution of vitamin D, fibrinogen and their ratio illustrates that the vitamin D and vitamin D/fibrinogen ratio is inversely related to the severity of ACS (median levels of Unstable angina > NSTEMI > STEMI), while fibrinogen is directly related to it. However, the median Fibrinogen levels of NSTEMI and STEMI groups are similar. Fig 1 illustrates the distribution of vitamin D, fibrinogen, and their ratio across ACS categories. A descending trend in vitamin D and the vitamin D/fibrinogen ratio, and an ascending trend in fibrinogen levels, are clearly evident from unstable angina to STEMI, reinforcing the statistical findings from the ANOVA and post-hoc comparisons.

The regression analysis (Multivariate and Binary logistic) showed that each 1nmol/L increase in vitamin D reduces CAD severity by 7.1%, with unstable angina being less severe and STEMI being the most severe one (P < .5). Similarly, vitamin D/fibrinogen ratio rise by 1 decreases the ACS severity by 6.2% (P < .5). Moreover, higher fibrinogen is associated with the

**Table 3. Comparison of vitamin D, fibrinogen, and vitamin D/fibrinogen ratio among ACS types.**

| ACS type | Vitamin D | | Fibrinogen | | Vitamin D/fibrinogen ratio | |
|---|---|---|---|---|---|---|
| | Mean rank | P value | Mean rank | P value | Mean rank | P value |
| Unstable Angina | 227.12 | .001 | 78.41 | .001 | 243.64 | .001 |
| NSTEMI | 151.95 | | 162.10 | | 147.55 | |
| STEMI | 110.45 | | 165.22 | | 110.16 | |

**Table 4. Correlation analysis between vitamin D, fibrinogen and vitamin D/fibrinogen ratio and ACS types.**

| Spearman Rho Test | | | Fisher Z test | | |
|---|---|---|---|---|---|
| Variable | Spearman's Rho | P-value | Variables | Z value | P value |
| Vitamin D (nmol/L) | −0.41 | .01 | Vitamin D VS Fibrinogen | 4.81 | .001 |
| Fibrinogen (g/L) | 0.25 | .01 | Fibrinogen VS Vitamin D/fibrinogen ratio | 0.34 | .73 |
| Vitamin D/fibrinogen ratio | −0.45 | .01 | Vitamin D VS Vitamin D/fibrinogen ratio | 5.15 | .001 |

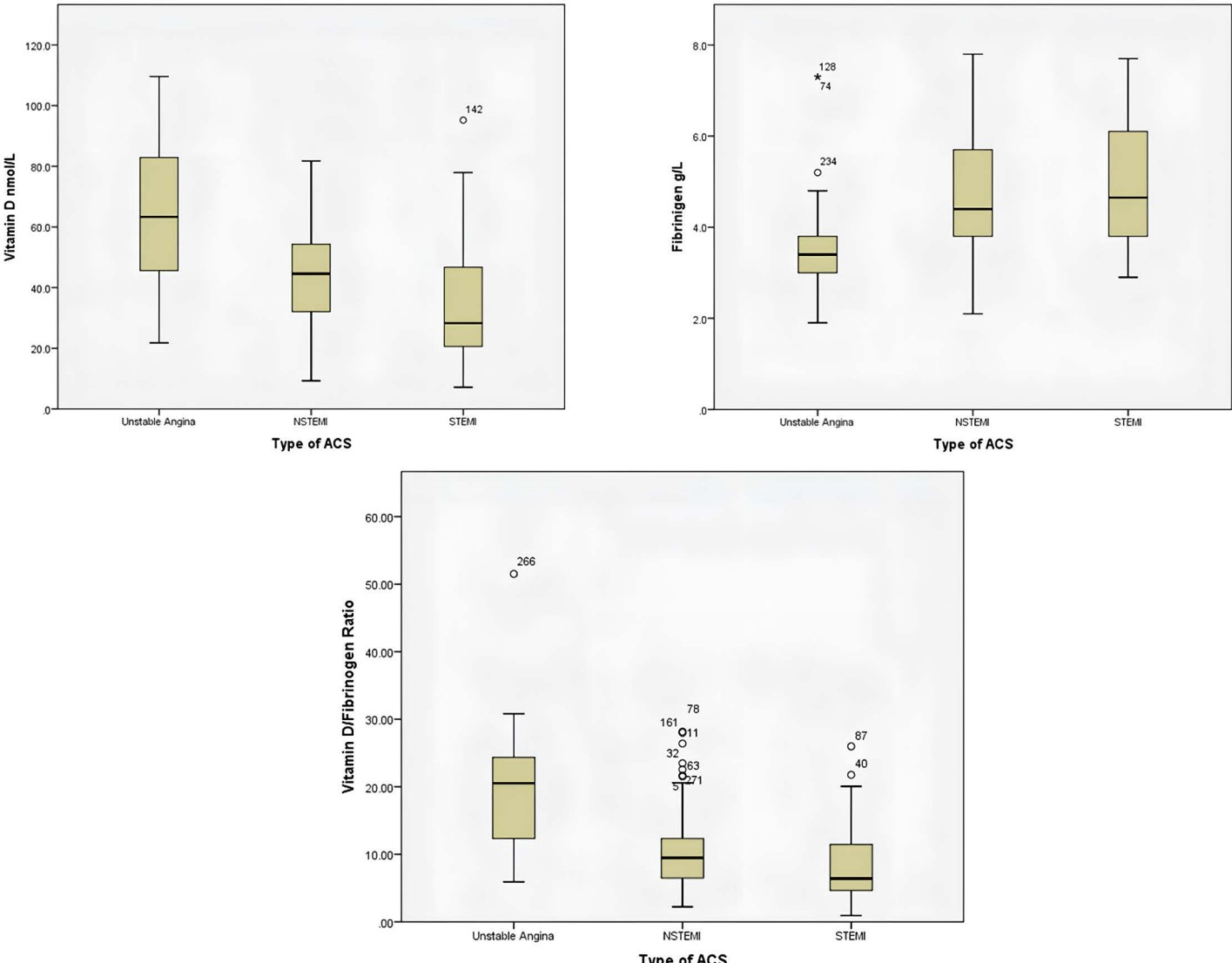

**Fig 1. Boxplot showing the distribution of serum vitamin D, fibrinogen, and vitamin D/fibrinogen ratio across ACS subtypes.** The median and interquartile range of serum vitamin D (nmol/L), and the vitamin D/fibrinogen ratio show a clear inverse relationship with ACS severity (from Unstable angina to STEMI), while those of fibrinogen show a direct association with ACS severity, although the median levels in the NSTEMI and STEMI groups are almost similar.

disease severity of ACS but significance is borderline (P = .06). The analysis explains 69% of the ACS variances accurately with Negelkerke $R^2 = 0.69$. The statistics are summarized in Table 5.

Collinearity diagnostics using Variance Inflation Factor (VIF) showed values of 3.04, 1.56, and 1.03 for vitamin D and vitamin D/fibrinogen ratio, fibrinogen and vitamin D/fibrinogen ratio, and vitamin D to fibrinogen, respectively. All these values are less than 5, indicating no significant multicollinearity among these variables (p < .05 for all).

## Discussion

The findings of this study align with the pathophysiological understanding that vitamin D deficiency and elevated fibrinogen contribute to thrombo-inflammatory processes in coronary artery disease and acute coronary events.

**Table 5. Logistic regression analysis (multivariate and binary).**

| Variable | Expo (B)/OR | 95% CI | | P value |
| --- | --- | --- | --- | --- |
| | | Lower | Upper | |
| **Age** | .978 | .937 | 1.020 | .292 |
| **Male** | 1.15 | .89 | 1.48 | .221 |
| **smoking** | .869 | .271 | 2.782 | .813 |
| **Diabetes Mellitus** | 1.920 | .658 | 5.600 | .232 |
| **Hypertension** | .657 | .197 | 2.196 | .495 |
| **Dyslipidemia** | 1.501 | .449 | 5.014 | .509 |
| **Vitamin D (nmol/L)** | .929 | .862 | .998 | .043 |
| **Fibrinogen (G/L)** | 2.786 | .940 | 8.254 | .064 |
| **Vitamin D/fibrinogen ratio** | .938 | .670 | 1.257 | .048 |

## 1. Vitamin D and acute coronary syndrome (ACS)

In recent years, beyond its classical role in bone metabolism, vitamin D has emerged as a promotor of cardiovascular health through its multidimensional anti-inflammatory, anti-thrombotic and endothelial-protective effects. Hence, its deficiency may not only reflect systemic inflammation but could also contribute to atherogenesis and plaque instability and thus to an increased cardiovascular risk. Wang et al in their study on the effects of vitamin D deficiency on cardiovascular disease concluded that vitamin D deficiency is associated with increased cardiovascular risk [16]. In one of their studies regarding vitamin D supplementation in cardiovascular diseases, Vacek and colleagues found a significant association between vitamin D deficiency and cardiovascular-related diseases, including hypertension, coronary artery disease, cardiomyopathy, and diabetes (all p < 0.05) [17]. These studies provided insights into vitamin D's role in cardiovascular diseases in general, but they did not address its role in ACS and its types in specific. In our study, we observe a significant inverse relation between vitamin D and ACS severity, being highest in unstable angina and lowest in STEMI patients, in the form of a clear gradient in mean values. These results are consistent with the previous studies on the subject. In their case-control study on vitamin D levels in myocardial infarction (MI) patients and non-MI patients, Giovannucci et al found that vitamin D levels were significantly lower in the MI group compared to the control group [18]. However, they did not study the effect of vitamin D on ACS types and severity separately. Pilz and colleagues in their study on the relationship between vitamin D and myocardial disease severity and related mortality concluded that lower vitamin D levels are associated with severe myocardial dysfunction and higher mortality, although they did not further study the inverse relation of vitamin D and ACS severity on the basis of ACS types [19]. This inverse relation between vitamin D and ACS severity suggests that vitamin D deficiency may enhance the inflammatory and thrombotic process, which is a key factor in ACS pathogenesis and severity [20]. These findings suggest vitamin D's potential as a modifiable contributor rather than a passive marker of ACS severity. In this context, future studies should explore the effects of vitamin D repletion strategies on the disease severity.

## 2. Fibrinogen and acute coronary syndrome (ACS)

Fibrinogen, being the ultimate member of the coagulation pathway, is the key factor in the pathogenesis of ACS [21]. Beyond its role in coagulation, fibrinogen is also a key factor in inflammation and endothelial injury. The direct association between fibrinogen and ACS severity reinforces its role in propagating occlusive thrombi, particularly in STEMI, where dense, fibrin-rich clots predominate. Our study observed a direct relationship between fibrinogen levels and ACS severity, with an incremental increase in mean fibrinogen level with the severity of ACS (STEMI > NSTEMI > Unstable angina). Previous research on the subject has shown the association of high fibrinogen levels with coronary artery disease (CAD), its

severity and higher mortality. In a meta-analysis on fibrinogen levels as a biomarker of outcomes in CAD patients, Cui et al found that fibrinogen level is associated with increased adverse events (ACS and others) and mortality in CAD patients; however, they did not study the relation between fibrinogen level with the type of ACS [22]. Xie et al, in their study related to fibrinogen levels in end-stage renal disease patients, concluded that higher fibrinogen levels combined with diabetes are associated with higher ACS-related mortality in these patients [23]. They, however, did not study the relationship with the ACS types individually. The pivotal role of fibrinogen in the process of thrombosis clarifies its direct relation with the severity of ACS as observed in our study.

## 3. Vitamin D/fibrinogen ratio and acute coronary syndrome (ACS)

One of the novel aspects of this study is the combined effect of both vitamin D and fibrinogen in the form of their ratio on ACS types. The vitamin D/fibrinogen ratio conceptually integrates two biologically antagonistic processes – anti-inflammatory protection versus prothrombotic activation – into a single measurable index. This contrasting effect of vitamin D and fibrinogen provided the rationale for this approach, since their ratio may provide the net thrombo-inflammatory state and prove a potential biomarker for ACS types and severity. Our study findings suggest a stronger inverse relation of the vitamin D/fibrinogen ratio with the ACS severity than that of vitamin D and fibrinogen alone (Spearman Rho 0.45 VS 0.41 VS 0.25). A similar relationship of vitamin D and fibrinogen with the burden and density of a clot was reported by Rautenbach and his colleagues [24]. In this study, they found that lower vitamin D and higher γ' fibrinogen levels are associated with denser clots with more adverse outcomes and vice versa. These findings strengthen the evidence in favour of our hypothesis, as clots in STEMI are the most stable clots as compared to unstable angina and NSTEMI.

Although this study shows a stronger correlation of the vitamin D/fibrinogen ratio to ACS severity, it is not statistically stronger than vitamin D alone, with an insignificant Fischer Z value (P = .73). However, the trend suggests that it may provide additional prognostic value as evident by the regression analysis. It is important to acknowledge that although statistical methods such as multivariable regression were employed to control for known confounders, the absence of random sampling may introduce unmeasured biases and generalizability. Future studies using probability-based or stratified sampling approaches could yield more precise population-level inferences.

## 4. Clinical implications

The clinical implications of this study could serve as both prognostic and therapeutic aspects of ACS. The inverse relation of vitamin D and the direct relation of fibrinogen could serve as prognostic biomarkers of ACS severity. Given their low cost, wide availability, and established use in other clinical settings, vitamin D and fibrinogen testing may serve as practical adjunctive tools for early risk stratification in ACS, especially when used alongside conventional markers like troponin and CK-MB. Secondly, these values could guide regarding the possible vitamin D supplementation and fibrinogen-lowering therapies to prevent and minimize the severity of ACS, as studied previously on the same subject showed mixed evidence [25,26]. While following the results of the Vitamin D and Omega 3 supplementation (VITAL) trial, Manson and colleagues found no significant improvement in cardiovascular disease end-points, but this trial involved healthy participants and not the established CAD or ACS or vitamin D deficient patients [27,28]. In our study, the mean vitamin D level in the STEMI group (35.59 ± 19.84 nmol/L) and NSTEMI group (43.24 ± 15.44 nmol/L) suggests vitamin D deficiency (vit D < 50 nmol/L), pointing towards the need for routine screening and potential supplementation in high-risk cardiovascular patients.

In this study, the vitamin D/fibrinogen ratio as a novel biomarker of ACS severity also showed a stronger inverse correlation with ACS severity as compared to vitamin D alone. The vitamin D/fibrinogen ratio, by integrating an anti-inflammatory/antithrombotic marker with a prothrombotic/pro-inflammatory marker, offers a more comprehensive and composite view of the thrombo-inflammatory phenomenon in ACS. Although the ratio's superiority over vitamin D alone

did not reach statistical significance in our cohort, the consistent trend and independent association with ACS severity suggest potential clinical utility. This ratio could complement existing biomarkers, including troponins and CRP. Troponins reflect myocardial injury, and CRP reflects systemic inflammation, while the vitamin D/fibrinogen ratio captures both inflammatory and coagulation pathways. In practice, since both vitamin D and fibrinogen are inexpensive, widely available tests, calculating this ratio could feasibly be integrated into ACS triage panels. In patients with equivocal troponin or CRP results, a low ratio may prompt closer monitoring, expedited intervention, or consideration of targeted therapies such as vitamin D supplementation or fibrinogen-lowering strategies. However, further research with a larger cohort could possibly strengthen this association, evaluating the feasibility of incorporating this parameter into ACS triage algorithms.

## 5. Strengths and limitations

The major strength of this study is the multinational study design, which adds to its reliability and generalizability. The robust methodology and comprehensive statistical analysis of the data also strengthen the credibility of this study. The limitations of this study are a relatively smaller cohort and a lack of longitudinal data on the long-term prognostic value of the biomarkers and effects of therapeutic interventions. Larger prospective studies are needed to validate these results before clinical implementation. Another limitation of the study is the lack of information regarding the seasonality, sun-exposure and sub-clinical inflammation that might have affected the results and should be considered while performing future studies in this context.

## Conclusion

In conclusion, this study provides valuable insights into the relationship between vitamin D, fibrinogen, and their ratio with ACS subtypes. The findings suggest that vitamin D and fibrinogen have contrasting effects on ACS risk, with vitamin D exerting protective effects and fibrinogen promoting atherogenesis and thrombosis. The introduction of the vitamin D/fibrinogen ratio as a potential biomarker for ACS risk stratification is a novel aspect of this study. Although its correlation to the ACS subtypes is not statistically stronger than that of vitamin D alone,our findings suggest that this ratio may be a useful tool for identifying patients at increased risk of ACS. Although novel, these findings require cautious interpretation due to the mentioned limitations. Further research is needed to validate the utility of this ratio in clinical practice and to determine whether interventions aimed at improving the vitamin D/fibrinogen ratio can reduce cardiovascular risk.

## Supporting information

**S1. Anonymized Data file.**
(XLSX)

## Acknowledgments

The authors would like to thank the deanship of scientific research at Shaqra University for supporting this work.

## Author contributions

**Conceptualization:** Himayat Ullah, Sarwat Huma.

**Writing – original draft:** Himayat Ullah, Hossam Shabana.

**Writing – review & editing:** Muhammad Ashraf, Nafisa Tahir, Ghulam Yasin, Mohammed Yunus, Mohamed Ahmed Muharram, Abdulrahman H. Shalaby, Ahmed Ali Hassan Ali, Mohamed Elwan Mohamed Mahmoud, Ahmed Mahrous Ahmed Ibrahim, Ahmed Farag Abd Elkader Elbwab, Ahmed Mohamed Ewis Alhawy, Ahmed Ahmed Mohamed Abotaha, Mahmoud Ezzat Abdelraouf, Mohammed S. Imam, Hossam Aladl Aladl Aladl, Taiseer Ahmed Shawky, Ashraf

Mohammed Said, Mahmoud Saeed Mahmoud, Kazem Mohamed Tayee, Reda Fakhry Mohamed, Ali Hosni Farahat, Mohammad Mossaad Abd Allah Alsayyad, Hesham El Sayed Lashin, Essam Yehia Ali Aggour, Hazem Sayed Ahmed Ayoub, Ayman Mohamed Salem Ahmed, Marwan Sayed Mohamed Ahmed, Abdelrahman E Metwally, Ahmed Saeed Abdelaziz, Tamer Ahmed Fouad Mohammed Hassan.

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
