## [Decision Letter · Decision Letter 0]

27 May 2025

Dear Dr. Yunus,

Thank you for submitting your manuscript to PLOS ONE. After careful consideration, we feel that it has merit but does not fully meet PLOS ONE’s publication criteria as it currently stands. Therefore, we invite you to submit a revised version of the manuscript that addresses the points raised during the review process.

We look forward to receiving your revised manuscript.

Kind regards,

Li Yang, M.D.

Academic Editor

PLOS ONE

**Journal Requirements:**

1. When submitting your revision, we need you to address these additional requirements. Please ensure that your manuscript meets PLOS ONE's style requirements, including those for file naming. The PLOS ONE style templates can be found at https://journals.plos.org/plosone/s/file?id=wjVg/PLOSOne_formatting_sample_main_body.pdf and https://journals.plos.org/plosone/s/file?id=ba62/PLOSOne_formatting_sample_title_authors_affiliations.pdf 2. Thank you for stating the following in the Acknowledgments Section of your manuscript: The authors would like to thank the deanship of scientific research at Shaqra University for supporting this work. We note that you have provided funding information that is not currently declared in your Funding Statement. However, funding information should not appear in the Acknowledgments section or other areas of your manuscript. We will only publish funding information present in the Funding Statement section of the online submission form. Please remove any funding-related text from the manuscript and let us know how you would like to update your Funding Statement. Currently, your Funding Statement reads as follows: The author(s) received no specific funding for this work.  Please include your amended statements within your cover letter; we will change the online submission form on your behalf. 3. In the online submission form, you indicated that “This will be provided on the authorized request to corresponding author”.  All PLOS journals now require all data underlying the findings described in their manuscript to be freely available to other researchers, either a. In a public repository, b. Within the manuscript itself, or c. Uploaded as supplementary information.This policy applies to all data except where public deposition would breach compliance with the protocol approved by your research ethics board. If your data cannot be made publicly available for ethical or legal reasons (e.g., public availability would compromise patient privacy), please explain your reasons on resubmission and your exemption request will be escalated for approval. 4. PLOS requires an ORCID iD for the corresponding author in Editorial Manager on papers submitted after December 6th, 2016. Please ensure that you have an ORCID iD and that it is validated in Editorial Manager. To do this, go to ‘Update my Information’ (in the upper left-hand corner of the main menu), and click on the Fetch/Validate link next to the ORCID field. This will take you to the ORCID site and allow you to create a new iD or authenticate a pre-existing iD in Editorial Manager. 5. Your ethics statement should only appear in the Methods section of your manuscript. If your ethics statement is written in any section besides the Methods, please delete it from any other section.

Reviewers' comments:

Reviewer's Responses to Questions

**Comments to the Author**

1. Is the manuscript technically sound, and do the data support the conclusions?

Reviewer #1: Partly

Reviewer #2: No

Reviewer #3: Partly

Reviewer #4: Yes

Reviewer #5: No

Reviewer #6: Yes

2. Has the statistical analysis been performed appropriately and rigorously?

Reviewer #1: Yes

Reviewer #2: Yes

Reviewer #3: No

Reviewer #4: Yes

Reviewer #5: No

Reviewer #6: Yes

3. Have the authors made all data underlying the findings in their manuscript fully available?

Reviewer #1: No

Reviewer #2: Yes

Reviewer #3: Yes

Reviewer #4: No

Reviewer #5: Yes

Reviewer #6: No

4. Is the manuscript presented in an intelligible fashion and written in standard English?

Reviewer #1: Yes

Reviewer #2: No

Reviewer #3: Yes

Reviewer #4: Yes

Reviewer #5: No

Reviewer #6: Yes

**Reviewer #1: ** Overall, the association between vitamin D, fibrinogen, and their ratio with acute coronary syndrome (ACS), which is innovative and clinically significant, however there is still room for improvement in research methods, data presentation and discussion.

While the study's use of non-probability convenience sampling achieved baseline feasibility, this methodological approach introduces inherent selection bias that may compromise the external validity of findings. It is recommended that subsequent studies adopt random sampling or other more scientific sampling methods to ensure the representativeness of the samples.

The sample size of 300 subjects demonstrates borderline adequacy for investigating complex biomarker interactions in ACS pathogenesis, potentially resulting in diminished statistical power to detect clinically meaningful associations.. It is possible to consider expanding the sample size for more in-depth analysis. The selection of statistical methods is basically reasonable, but given the limitations of the sampling method, its impact on the statistical results should be further evaluated in the discussion of the results. At the same time, in-depth analysis of the clinical significance of the statistical results can be increased.

Although the discussion part compares the research results with previous studies, The discussion would benefit from expanded consideration of translational applications for the vitamin D/fibrinogen ratio, particularly concrete recommendations for clinical implementation and biomarker-guided therapeutic strategies.

The current conclusions are supported by certain data, but they need to be interpreted with caution given the limitations of the study. It is recommended to more clearly point out the impact of the limitations of the study on the generalizability of the conclusions in the conclusion. Some content is repeated, such as the repeated mention of the research background in the introduction and discussion. The content can be streamlined to make the discussion clearer. Certain complex sentence structures may impede readability and could be streamlined for improved clarity.

While ethical approval was obtained, enhanced documentation of patient privacy safeguards would strengthen methodological transparency. Furthermore, future investigations should examine potential ethnic and geographic variations in biomarker relationships to enhance the generalizability of findings across diverse populations.

**Reviewer #2: ** I have read this article in detail. This article mainly discusses the correlation between vitamin D/fibrinogen and different types of acute coronary syndrome (ACS). In the currently published articles, there is a lack of data on Arab patients in the Middle East, and this article effectively fills this gap.

Unfortunately, there are still some issues that need to be addressed in this article.

1. Grammar and article structure

I have to regretfully point out that the grammar and structure of this article are confusing. There is a significant difference in the quality of grammar in the article. In some parts, such as the beginning of the article, the language expression of the article has not reached the level of current mainstream articles. In addition, the production of tables and images is also quite rough. The position of the image is incomprehensible and placed at the end. The low resolution of the image makes it difficult to read.

2. References cited

The citation level of this article does not support the author's viewpoint. It is recommended to choose updated, more mainstream, and frequently cited articles when selecting references. And in terms of article type, it is recommended to choose more basic experimental or clinical papers rather than reviews. Additionally, there is a lack of references to the latest guidelines currently available.

3. Target selection

At the beginning of the article, the author did not provide a clear explanation for why the ratio of vitamin D to fibrinogen was chosen. This can cause confusion for readers. In the following content, the author claims that STEMI is more dangerous than NSTEMI. This seriously affects the conclusion of this article. In the 2017 ESC NSTEMI guidelines, it is recommended to use GRACE to evaluate the degree of danger. In this article, there is a lack of evaluation of NSTEMI, which makes the conclusion completely unsupported. From the logic of the article, the changes in vitamin D/fibrinogen are more like manifestations of ACS classification rather than the severity of ACS

**Reviewer #3:**  Major Comment:

While the study presents interesting statistical associations between vitamin D, fibrinogen, and ACS severity, it lacks discussion on potential mechanistic pathways or how these findings might translate into clinical decision-making. The authors should elaborate in the Discussion on the biological plausibility of the vitamin D/fibrinogen ratio affecting ACS pathophysiology and address how this biomarker could be incorporated into existing clinical workflows for risk stratification or therapeutic monitoring.

Minor Comments:

1. Overextended Introduction

The Introduction is overly long and includes detailed explanations that are more appropriate for the Discussion section, such as the rationale for comparing biomarkers and interpretation of their roles. The authors are encouraged to streamline the introduction to focus on background, knowledge gaps, and objectives, moving interpretive content to the Discussion.

2. Patient Selection Criteria Should Be Visualized

In the Methods section, the inclusion and exclusion criteria are clearly described in text, but a flowchart (e.g., a CONSORT-style diagram) would greatly enhance clarity and transparency, particularly in a multicenter study.

3. Clarify Country-Level Differences

The results mention no significant differences across countries, but no further stratified analysis or sensitivity analysis is provided. A brief comment on inter-country consistency or variability (e.g., healthcare systems, diagnostic protocols) would strengthen the generalizability claim.

4. Typographical and Formatting Issues

Several instances of awkward phrasing (e.g., “To make things simple...”) and inconsistent formatting (e.g., line breaks, inconsistent p-value notation) are present throughout the manuscript. A thorough copyedit is recommended to improve readability and professionalism.

5. Clarify Ratio Interpretation

The interpretation of the vitamin D/fibrinogen ratio as a “stronger” correlate than vitamin D alone is not statistically supported (Fisher Z-test p = 0.73). This should be clearly stated in both Results and Conclusion to avoid overstating the findings.

**Reviewer #4: ** �1�Does Table 1 need to explain anything? Table 1 is just common clinical knowledge and doesn't seem to contribute to the paper.

2�What is the economic value of measuring vitamin D and fibrinogen in the assessment of acute myocardial infarction? What are the advantages over troponin and CK-MB? It seems to have little clinical significance.

3�The research has certain advantages in using data from multiple countries, but the sample size used in the study is too small, resulting in weak proof strength.

**Reviewer #5: ** This study is of particular interest as it explores the potential of identifying novel biomarkers in the correlation between serum vitamin D levels, fibrinogen, and their ratios, and the severity of acute coronary syndromes. These syndromes include unstable angina, non-ST-elevation myocardial infarction, and ST-elevation myocardial infarction.

The fact that the study was conducted in collaboration with other centers suggests that geographic diversity and generalizability were also taken into consideration.

The hypothesis that the vitamin D/fibrinogen ratio functions as a biomarker is both novel and clinically beneficial. However, given the prevailing circumstances, it is our conviction that the publication system must undergo enhancement to establish a nexus with scientific evidence and clinical applications.

It is imperative that the following points be given due consideration, and that they be linked to the publication.

Major Points

1. Overstatement of Conclusions

The vitamin D/fibrinogen ratio demonstrated a slightly stronger negative correlation with ACS severity compared to vitamin D alone (Spearman’s ρ = -0.45 vs. -0.41); however, this difference was not statistically significant (Fisher Z = 0.34, p = 0.73). The manuscript currently presents this ratio as if it is a superior biomarker. Although the numerical difference may appear clinically meaningful, the lack of statistical significance indicates that this claim should be moderated. Given the absence of statistical significance, the authors should emphasize the exploratory nature of this finding and clearly state the need for further validation.

2. Potential Bias Not Adequately Addressed

The logistic regression model does not appear to account for key potential confounding factors that could affect serum vitamin D and fibrinogen levels, such as seasonal variation, sunlight exposure, renal function, vitamin D supplementation, and chronic inflammatory conditions. If this is a multicenter study, addressing potential bias is particularly critical to maintain analytical rigor. The authors should clarify whether these factors were assessed, or explicitly acknowledge their absence as a limitation of the study.

3. Timing of Sample Collection

The timing of serum sample collection in relation to symptom onset or hospital admission is not clearly stated. Considering that fibrinogen is an acute-phase reactant, this timing is critical to properly evaluate its utility as a biomarker. The authors should clarify the timing of sample collection and consider including this aspect in their analysis or discussion.

4. Clinical Interpretation of Vitamin D Levels

While the mean serum vitamin D levels are reported, there is no mention of clinically relevant thresholds (e.g., deficiency <25 nmol/L, insufficiency <50 nmol/L, sufficiency >75 nmol/L) or the number of patients falling into each category. From the perspective of evaluating its utility as a clinical biomarker, the authors are encouraged to analyze and present this information accordingly.

Minor point

1. Image Quality of Figure 1

The resolution of Figure 1 is too low to be legible. Please provide a high-resolution version to ensure that readers can accurately interpret the figure.

2. Consistency of Terminology

To avoid reader confusion and improve readability, all abbreviations (e.g., STEMI, NSTEMI, UA, ACS) should be spelled out in full at first mention, followed by the abbreviation in parentheses. Please ensure this is applied consistently throughout the manuscript.

3. Language and Grammar

To enhance the overall readability of the manuscript, please carefully review the English writing and consider the following points:

• Avoidance of Redundant Expressions

Example:

“Vitamin D levels were significantly lower (p < 0.001) and fibrinogen levels significantly higher (p < 0.001)…”

→ The word “significantly” is unnecessarily repeated. This can be revised for conciseness.

• Avoidance of Colloquial Expressions

Example:

“To make things simple…”

→ This phrasing is too casual for a scientific article. Please replace it with more formal language.

• Correction of Grammatical Errors and Awkward Phrasing

Example:

“…explaining the whole process, including the study benefits to them and the community, the possible risks to them, and the confidentiality and anonymization of their data, to them.”

→ The repetition of “to them” makes the sentence unnecessarily redundant.

• Avoidance of Repetitive Word Usage

Example:

“The strength of the correlation (Spearman Rho) showed a significant correlation…”

→ The word “correlation” is unnecessarily repeated. Please revise for clarity.

• Simplification of Overly Long and Complex Sentences

In particular, the Methods and Discussion sections contain several overly long and complex sentences, which may reduce readability. Please consider breaking them into shorter, more direct sentences.

4. Reconsideration of Cited Literature

Several of the key references related to vitamin D and cardiovascular disease are over a decade old. Considering the ongoing research in this field, it is recommended to include more recent and high-quality studies, such as randomized controlled trials or meta-analyses (e.g., the VITAL trial). Updating the references will strengthen the scientific relevance of the manuscript.

**Reviewer #6:**  Point 1: Lack of systematic approach: As stated in the study, the vitamin D/fibrinogen ratio appears an ad hoc metric. Strengthen the rationale by citing pathophysiology.

Suggestion: Please outline how low vitamin D contributes to endothelial dysfunction and high fibrinogen to thrombosis in ACS, then explain why their ratio would reflect net thrombo‐protective status.

Point 2: Power calculation mismatch: The methods claim a required sample size of 60 (4% population prevalence), yet you enrolled 300 six times that number??

Suggestion : Recalculate the power analysis using effect sizes from prior studies of vitamin D and fibrinogen in ACS to demonstrate that n = 300 yields >80% power to detect clinically meaningful differences across three groups.

Point 3: This study has multiple pairwise comparisons (UA vs. NSTEMI vs. STEMI) without adjustment.

Suggestion: Try to apply a correction (Bonferroni or Holm) to control familywise error, and report both raw and adjusted p-values.

**Do you want your identity to be public for this peer review?** For information about this choice, including consent withdrawal, please see our Privacy Policy

Reviewer #1: No

Reviewer #2: No

Reviewer #3: No

Reviewer #4: No

Reviewer #5: No

Reviewer #6: **Yes: ** Dr. Arindam Chatterjee

---

## [Author Response · Author response to Decision Letter 1]

5 Jun 2025

Point by point response to Reviewer 1:

Thanks for the valuable comments and input into the improvement of the manuscript and really appreciate your kind words regarding the study.

Regarding the sampling technique, we totally agree that random sampling is the ideal technique, but the non-probability convenience sampling was more practical choice due to the emergency nature of ACS and the access constraints in three different international centers, having different patient record management systems. However, we have added this to the limitations section in the revised manuscript.

Regarding the sample size of 300 patients, we agree that larger studies would increase statistical power and external validity, our sample well exceeded the minimum calculated threshold of 60 patients. Moreover, we acknowledge this and have added a note encouraging future larger-scale studies.

Regarding the effects of sampling on statistical results, we have included a brief discussion of how the non-probability design may have influenced the statistical estimates, and what cautions have been done for analysing the effect sizes and confidence intervals with marginal p values in the discussion of the results.

Regarding the translational application of the vitD/fibrinogen ratio, the discussion is further strengthened by clinical implication of the same in the revised manuscript as suggested. The revised manuscript now elaborates on how the ratio could be integrated into existing clinical workflows and risk scores in adjunct to troponins or CRP.

Regarding the Conclusion, the sentence is added as, “These findings, while novel, require cautious interpretation given methodological limitations and need further validation,” in order to give a more cautious statement based on the study limitations.

Regarding the repetition and redundancy in the introduction and discussion, the revised manuscript has removed several points like repeating introductory text from the Discussion describing the ACS subtypes.

The manuscript is revised with the help of professional copyediting and proofreading, making it simpler and easier to understand.

Regarding ethics and patient privacy, additional explanation has been added to Methodology section mentioning the anonymisation of the data and authorized access to the patient data.

Regarding the ethnic and geographic variations and generalizability, while we did stratify by country and found no statistically significant differences as mentioned Brown-Forsythe test , we have further emphasized its importance in the discussion.

Point by point response to Reviewer 2:

Thanks for your valuable input to the manuscript.

1. Regarding the Grammer and article structure we really appreciate your comments and the authors have revised the manuscript with the help of professional copyeditor and proofreader. The production of tables and figures are formatted according to the journal’s guidelines once article is accepted and sent for production. The figure will be placed where it is mentioned in the text, i.e., after table 4 and is prepared according to the journal guidelines having more than 300dpi. However, we have revised and improved the quality of the figure to 1200 dpi.

2. Regarding the cited references, 10 references are from the past 2021-2024, 7 from year 2010 to 2020, and 8 from 2001 to 2010, showing upto 65% of references are from the last 10 years and 40% from the last 3 years. None of the reference is older than 2001. Among these 12 are original clinical/observational studies, 3 metanalysis, 4 systematic reviews, 4 pathophysiologic and literature reviews and 2 are books chapters with guidelines, showing a healthy balance of the different components of scientific literature. These references are selected based on scientific relevance according to academic standards and not the citation count.

3. Regarding the comment, “At the beginning of the article, the author did not provide a clear explanation for why the ratio of vitamin D to fibrinogen was chosen”, I would humbly state that the first line of the abstract start with the statement, “There is emerging evidence suggesting that vitamin D and fibrinogen play contrasting roles in ACS pathophysiology and their combined impact, expressed as the vitamin D/fibrinogen ratio, can be a potential biomarker for ACS severity.” Similarly in the Introduction section, after the introduction of the ACS and the biomarkers in the question, the selection of their ratio is explained as, “These contrasting effects of vitamin D and serum fibrinogen may give rise to the hypothesis that the interplay of these two important biomarkers could affect the course and severity of CAD.” I hope this clarifies the rationale of using the vit D/fibrinogen ratio to the readers from the first line.

Regarding the comment, “the author claims that STEMI is more dangerous than NSTEMI. This seriously affects the conclusion of this article”, I would humbly state that it is not my claim, rather it is supported by published evidence as referenced in the manuscript and also supported by the pathophysiologic and vascular changes in STEMI as compared to NSTEMI (Table 1).

Regarding the comment, “In the 2017 ESC NSTEMI guidelines, it is recommended to use GRACE to evaluate the degree of danger. In this article, there is a lack of evaluation of NSTEMI, which makes the conclusion completely unsupported.” We agree that formal risk scores, such as GRACE or TIMI quantify the individual patient level severity, they do not capture the catogoric level severity of ACS sub-types. The GRACE score, which accounts for age, pulse rate, Systolic BP, creatinine, cardiac arrest, ST-segment deviation, abnormal cardiac enzymes and Killip class for the severity, is used for estimating the severity of individual ACS patients, and not the ACS sub-groups. However, even in the GRACE scoring system, the ST-segment deviation will always be positive for STEMI, but may be negative for NSTEMI, denoting the higher scores in STEMI Vs NSTEMI if the other components are equal. Having said that, none of the objectives of the study is related to the outcome of NSTEMI or any other type ACS; rather all the objectives are regarding the relation of vitamin D, fibrinogen and their ratio to the ACS sub-types.

Point by point response to Reviewer 3:

Major Comment:

Thanks for highlighting the importance of addressing mechanistic plausibility and clinical application of our findings. We respectfully state that both of these points are elaborated in the discussion section as we state,

“Their contrasting effect provided the rationale for this approach, since this ratio may prove as a potential biomarker for ACS types and severity… These findings strengthen the evidence in favour of our hypothesis, as clots in STEMI are the most stable clots as compared to unstable angina and NSTEMI.”

(Page ~17, Paragraph starting with “One of the novel aspects…)”. This paragraph provides the biological plausibility and mechanistic rationale to our hypothesis. In another paragraph in the “Clinical Implication” subsection of the Discussion, we state that,

“The inverse relation of vitamin D and the direct relation of fibrinogen could serve as prognostic biomarkers of ACS severity. Secondly, these values could guide regarding the possible vitamin D supplementation and fibrinogen lowering therapies…”

(Page ~18, Paragraph starting with “The clinical implications of this study…”).

Moreover, in the revised manuscript we have added a paragraph stating,

“In clinical practice, this may serve as a surrogate marker of severity and outcome, complementing the existing biomarkers such as troponin and CRP, particularly in patients where these traditional markers are equivocal.”

Minor Comment:

1. Regarding the overextended Introduction, it has been revised, now having just 736 words (including a table), highlighting the importance of the study, gaps in the literature, rationale behind the study and the main objectives.

2. Regarding the CONSORT type flow diagram we appreciate the reviewer’s suggestion, however, we respectfully believe that in the context of our observational, cross-sectional study, such a diagram is not essential, as it is not a clinical trial, having no drop out of the patients and whole of the cohort is included in the study, and the inclusion and exclusion criteria with sample size is clearly and very simply stated. Adding a diagram may cause redundancy, however, if the editor deems it necessary for consistency or presentation purposes, we would be happy to include a simplified flow diagram in the final version.

3. Regarding the inter-county variability, the statistical analysis showed no significant differences in the variables among the ACS categories of the three counties as we state in the results, “Brown-Forsythe test showed no significant difference among the patients from the three countries on the basis of vitamin D, fibrinogen and their ratio (P > .05 for all the three).” Based on this fact, the patients of all the countries can be considered and analyzed as a single cohort and commenting on the variability may go against the statistical proof.

4. Regarding typographical and formatting issues, we acknowledge this issue and the revised manuscript is assessed through professional copy-editing and proofreading to improve this shortcoming.

5. Regarding the vit D/fibrinogrn ratio interpretation, we acknowledge the suggestion and have added a clear note to the conclusion that the ratio is not significantly stronger than isolated vitamin D levels statistically.

Point by point response to Reviewer 4:

Thanks for the valuable input for improving the manuscript.

Regarding Table 1, we appreciate your comment, but we respectfully submit that Table 1 was included to simplify the differences of the ACS subtypes showing their severity to support the rationale for exploring the pro- and anti-thrombotic and pro- and anti-inflammatory biomarkers. This is to serve a pedagogical and contextual purpose for the readers. However, if considered redundant, we are open to either relocating the content to the supplementary material or removing the table entirely.

2. Regarding economic feasibility, both the tests are are inexpensive, widely available, already in use and their combination may offer insight into the inflammatory and thrombotic burden, which is not captured by troponin alone. This is especially relevant for early risk stratification, and possible treatment like, vitamin D supplementation. Moreover, we have added this point to the Discussion subsection clinical implication.

Regarding advantage over troponins and CKMB, our study does not aim to replace these markers but to explore adjunctive biomarkers that may provide additional prognostic or stratification value, particularly in borderline or intermediate-risk presentations.

3. Regarding sample size, we agree that a larger sample size would enhance the strength and generalizability of the findings. However, this study was designed as a preliminary, multicenter exploratory analysis, and the sample size of 300 well exceeded the minimum required sample to detect a moderate effect size based on prior biomarker studies. Having said that, we have also added this to the limitations that larger studies are needed to validate the results before clinical application.

Point by point response to Reviewer 5:

Thanks for your valuable input into the improvement of the manuscript.

Major Points:

1. Regarding overestimation of the conclusion based on the ratio, we appreciate and acknowledge this point and have clearly added this clarification of statistical non-superiority into the various sections of the manuscript. As we have added to the results the statement,

“However, although the correlation vitamin D/fibrinogen ratio to ACS types was stronger (0.45) as compared to that of vitamin D alone (0.41), it was not significantly stronger statistically (Fisher Z value 0.34, P value 0.73)…”

Similarly, we have added to the Discussion subsection Vitamin D/Fibrinogen ratio and ACS, as

“Although this study shows a stronger correlation of vitamin D/fibrinogen ratio to ACS severity, it is not statistically stronger than vitamin D alone with an insignificant Fischer Z value (P = .73).”

2. Regarding potential bias, we acknowledge your valuable observation. Although vitamin D supplementation, liver or kidney disease, malignancy, anticoagulants were the exclusion criteria mentioned in the methodology, the seasonality and sun-exposure was not addressed, and have been added as one of the limitations, stated as,

“Another limitation of the study is the lack of information regarding the seasonality, sun-exposure and sub-clinical inflammation that might have affected the results and should be considered while performing future studies in this context.”

3. Regarding the timing of sample collection, we appreciate your observation. Although previously not mentioned, we have added this to the methodology as all the blood samples are collected at the time of admission, within the first three hours.

4. Regarding clinical interpretation of Vitamin B levels among ACS groups, we really appreciate pointing out this omission. In the revised manuscript, we have added the latest reference of vitamin D levels to the introduction and added the status of vitamin D deficiency among different ACS groups of the cohort.

Minor point:

1. Regarding Figure quality, although we have added the prescribed dpi figure, in the revised version we have enhanced the dpi even more.

2. Regarding Consistency of terminology, we have reviewed the manuscript to ensure that all abbreviations (ACS, STEMI, NSTEMI, UA) are spelled out in full at first mention, followed by abbreviation in parentheses and used consistently throughout the text.

3. Regarding Grammer and language, we have revised the manuscript with the help of professional copyediting and proofreading, and all the mentioned error and omissions have been corrected.

4. Regarding citations, we really thank the suggestion to strengthen the manuscript by incorporating more recent and high-quality studies and we have now included the VITAL trial and its continuation which is a landmark randomized controlled trial evaluating the role of vitamin D supplementation in cardiovascular outcomes.

At the same time, we respectfully emphasize that several references that are more than a decade old were deliberately retained because they provide foundational insights into the biological mechanisms, epidemiological associations, and historical development of vitamin D and fibrinogen research in cardiovascular disease. These include seminal studies such as:

*Pilz et al. (2008) – A cornerstone observational study linking low vitamin D to cardiovascular mortality.

*Wang et al. (2008) – A key meta-analysis establishing the association between vitamin D status and incident cardiovascular events.

*Danesh et al. (2005) – A major pooled analysis on fibrinogen and vascular risk, still widely cited in biomarker literature.

Out of 28 citations only 7 are older than 2012, while most of them are older than 2015 i.e; upto 75%. The older references continue to be cited in recent systematic reviews and clinical guidelines due to their methodological robustness and ongoing relevance.

Therefore, while we have updated the manuscript with newer studies, we have retained select older references to maintain a comprehensive and historically contextualized discussion. All retained references were selected for their scientific relevance, not publication age.

Point by point response to Reviewer 6:

Point 1: We really appreciate this important point. While the rationale for using the vitamin D/fibrinogen ratio was discussed, we agree it can be strengthened by elaborating on the pathophysiological mechanisms linking both components to ACS. In this context we have added a paragraph to the introduction section elaborating the rationale further.

Point 2: The power analysis, using G*Power version 3.1 has been done, showing power of more than 97% for sample size of 300, showing clinically meaningful difference among the three ACS groups.

Point 3: Thanks for this valuable suggestion, and we have applied Bonferroni and Games-Howell as stated in the results as,

---

## [Decision Letter · Decision Letter 1]

3 Jul 2025

Dear Dr. Yunus,

Thank you for submitting your manuscript to PLOS ONE. After careful consideration, we feel that it has merit but does not fully meet PLOS ONE’s publication criteria as it currently stands. Therefore, we invite you to submit a revised version of the manuscript that addresses the points raised during the review process.

We look forward to receiving your revised manuscript.

Kind regards,

Li Yang, M.D.

Academic Editor

PLOS ONE

Journal Requirements:

Reviewers' comments:

Reviewer's Responses to Questions

**Comments to the Author**

Reviewer #1: (No Response)

Reviewer #2: (No Response)

Reviewer #3: (No Response)

Reviewer #5: All comments have been addressed

Reviewer #6: All comments have been addressed

2. Is the manuscript technically sound, and do the data support the conclusions?

Reviewer #1: Partly

Reviewer #2: Yes

Reviewer #3: (No Response)

Reviewer #5: Yes

Reviewer #6: Yes

3. Has the statistical analysis been performed appropriately and rigorously?

Reviewer #1: Yes

Reviewer #2: Yes

Reviewer #3: (No Response)

Reviewer #5: Yes

Reviewer #6: Yes

4. Have the authors made all data underlying the findings in their manuscript fully available?

Reviewer #1: No

Reviewer #2: Yes

Reviewer #3: (No Response)

Reviewer #5: Yes

Reviewer #6: Yes

5. Is the manuscript presented in an intelligible fashion and written in standard English?

Reviewer #1: Yes

Reviewer #2: Yes

Reviewer #3: (No Response)

Reviewer #5: Yes

Reviewer #6: Yes

Reviewer #1: The manuscript "Relationship of Vitamin D, Fibrinogen and Their Ratio with Acute Coronary Syndrome" has made substantial improvements since the prior review, particularly in addressing clinical implications, statistical power, and manuscript clarity. However, several key issues require further attention to ensure compliance with PLOS ONE standards and enhance the rigor of the study.

1. Data Availability and Transparency The current data availability statement is insufficient and vague, failing to meet PLOS ONE’s requirement for unrestricted data sharing.

2. Statistical Methods and Reporting� While multiple comparisons were corrected using Bonferroni and Games-Howell tests, the rationale for choosing these methods and their impact on results is unclear. Additionally, potential collinearity between vitamin D and the vitamin D/fibrinogen ratio was not assessed.

3. Research Design Limitations The non-probability convenience sampling introduces selection bias, but the discussion lacks sensitivity analysis or sub-group comparisons to assess its impact.

4. Mechanistic and Clinical Context The discussion of biological plausibility for the vitamin D/fibrinogen ratio remains superficial, and clinical application suggestions are not contextualized with practical challenges.

5. Language and Formatting: Minor grammatical errors, inconsistent terminology, and formatting issues persist, affecting readability.

6. Ethical and Compliance Details: While ethics approval is mentioned, the methods lack detail on patient consent processes and data anonymization measures.

The study provides valuable insights into novel biomarkers for ACS, but further refinements are necessary to address data transparency, statistical rigor, and clinical relevance. Once the above issues are resolved, the manuscript will be suitable for publication in PLOS ONE.

Reviewer #2: I am very pleased to see this revised manuscript. The quality of this manuscript has significantly improved, and the readability of the article has also significantly improved. The conclusion of the article is also consistent with the overall situation of this clinical study. But there are still some issues that we hope to improve before publication.

Firstly, the title of the discussion section lacks numbering. A reliable title can further enhance the readability of the overall article structure and clearly display the structure of the article. If you do not intend to change the font size of the main title and subtitle, it is recommended to add numbering.

Secondly, discuss the structure of the article in the discussion section. The discussion section does not simply list the sources of inspiration for this article. I hope to see further thinking and meaningful discussions. And why is content similar to a conclusion placed at the beginning of the discussion?

Reviewer #3: (No Response)

Reviewer #5: General comments:

This revised manuscript shows significant improvements in clarity, structure, and scientific rigor. The authors have thoroughly and clearly addressed all of the previous reviewers' comments. The study design is observational and based on convenience sampling, but it is appropriately justified in the context of a clinical setting and multi-center logistics.

Minor suggestions for improvement:

• Please confirm the consistency of terminology, particularly the standardized notation for the “vitamin D/fibrinogen ratio.”

• Figure 1 should be more appropriately integrated into the flow of the narrative in the results section.

• The authors should concisely emphasize that while the vitamin D/fibrinogen ratio is not statistically superior, its clinical utility as a composite marker is a promising area.

Reviewer #6: (No Response)

**Do you want your identity to be public for this peer review?** For information about this choice, including consent withdrawal, please see our Privacy Policy

Reviewer #1: No

Reviewer #2: **Yes: ** Zezhong Zhong

Reviewer #3: No

Reviewer #5: No

Reviewer #6: **Yes: ** Arindam Chatterjee

---

## [Author Response · Author response to Decision Letter 2]

5 Jul 2025

Response to reviewer #1:

We really appreciate valuable comments towards the improvement of the manuscript. We are thankful to your kind words and appreciation of the authors’ hard work.

1. Regarding data availability and transparency, we have added a statement regarding the data availability according to PlosOne guidelines and attached a supplementary file.

2. Regarding point 2, we would clarify that we have mentioned the rationale behind using Bonferroni and Games-Howell post-hoc tests is the Statistical Analysis section, stating;

“Analysis of variance (ANOVA) followed by the Kruskal-Wallis test was performed to show the difference among the ACS groups. The Games-Howell post hoc was performed to evaluate the inter-group pairwise differences in Vitamin D, Fibrinogen levels, and the VitD/Fibrinogen ratio (Unstable angina VS NSTEMI VS STEMI)”.

However, to improve upon this, we have revised and rewrote it as,

“Bonferroni post-hoc tests were used for pairwise comparisons where homogeneity of variance was assumed (as tested using Levene’s test), while Games-Howell was employed where variances were unequal. This approach ensures rigorous control over Type I error in both parametric and non-parametric contexts.”

Regarding collinearity, we performed collinearity diagnostics using Variance Inflation Factor (VIF) and the statistics are added to the results as, “Collinearity diagnostics using Variance Inflation Factor (VIF) showed values of 3.04, 1.56, and 1.03 for vitamin D and vitamin D/fibrinogen ratio, fibrinogen and vitamin D/fibrinogen ratio, and vitamin D to fibrinogen, respectively. All these values are less than 5, indicating no significant multicollinearity (p < .05 for all)”.

3. Regarding sensitivity analysis, we have performed the sensitivity analysis in the revised version and mentioned in the results as,

“To further mitigate the risk of selection bias, stratified sensitivity analyses across the three study sites were also performed, which showed consistent trends in biomarker levels and their associations with ACS severity across the three countries”.

4. Regarding the mechanistic, the discussion on the biological plausibility of vitamin D/fibrinogen ratio has been discussed in details in the introduction section, by first discussing the roles of vitamin D and fibrinogen separately, then discussing their contrasting effects and then stating, “Although the effects of vitamin D and fibrinogen on vascular health and coronary arteries have been studied individually, their combined effect has never been studied previously [15]. These contrasting effects of vitamin D and serum fibrinogen may give rise to the hypothesis that the interplay of these two important biomarkers could affect the course and severity of CAD. Vitamin D deficiency and high serum fibrinogen levels may play a synergistic role in coronary vascular inflammation, endothelial dysfunction, platelet aggregation and atherosclerosis, thus adversely affecting the ACS severity.”

Restating all these will not only lengthen the discussion but also cause redundancy. However, we have further explained the clinical applicability and practical challenges of the ratio in the end of clinical implications subsection as,

“Clinically, however, implementation challenges include assay availability, inter-laboratory variation, and lack of standardized cutoffs. These barriers must be addressed before routine use in ACS triage.”

5. Regarding minor language and formatting issues, we are really thankful to point this drawback out. We have used a professional proofreader and copy-editor has been used to improve the revised manuscript.

6. Regarding detail on patient consent processes and data anonymization measures, it has already been mentioned in the Material and methods section as,

“A written informed consent was taken from each patient explaining the study process, including the potential benefits to participants and the community, possible risks, and measures to ensure confidentiality and data anonymization.”

Once again, thank you for appreciating our efforts and providing valuable insights and input in the improvement of the manuscript.

Response to reviewer 2:

We are really thankful for your encouragement and valuable comments.

Regarding numbering and font size, we totally agree with your valuable suggestion. Since the font size is in accordance with the journal’s suggested template, we have added numbering to the Discussion sections. This will enhance the clarity and navigation.

Regarding suggestion for a meaningful discussion, we really appreciate your insightful analysis. We have revised the discussion and added deeper mechanistic of vitamin D, fibrinogen and their ratio into the respective subsections in the discussion, keeping in view the word limits and redundancy. We hope that this will add to the meaningful discussion.

Regarding the opening paragraph of the discussion, we agree that it is resembling conclusion and we have restructured it according and now discussion starts with the key findings of this study in context with previous literature on the pathophysiologic processes of vitamin D and fibrinogen in CAD.

We are really thankful for your constructive suggestions. We believe these changes have substantially improved the clarity, structure, and depth of the discussion.

Response to reviewer 5:

We are really thankful to you for the encouraging feedback and acknowledging the revisions made to improve the manuscript’s clarity, structure, and methodological transparency. We are grateful for the constructive suggestions and have addressed them in the revised version one by one.

Regarding consistency of the terminology, we have reviewed and confirmed that uniform terminology is used throughout the manuscript. Any alternative phrases have been replaced by uniform standard phrasing, for example vitamin D/fibrinogen ratio.

Regarding the narration of Figure 1 in the results section, we agree with your suggestion and added the narration before the figure as,

“Figure 1 illustrates the distribution of vitamin D, fibrinogen, and their ratio across ACS categories. A descending trend in vitamin D and the vitamin D/fibrinogen ratio, and an ascending trend in fibrinogen levels, are clearly evident from unstable angina to STEMI, reinforcing the statistical findings from the ANOVA and post-hoc comparisons.”

Regarding the emphasis on the statistical non-superiority of the ratio over vitamin D alone, we totally agree to your suggestion and an emphasis is made on this in the discussion section, adding,

“Although this superiority is not statistically significant to vitamin D alone, its dual representation of anti-inflammatory and prothrombotic axes may enhance clinical relevance as a composite risk marker in ACS assessment”.

We believe these refinements have further improved the clarity, scientific integrity, and clinical value of our work.

---

## [Decision Letter · Decision Letter 2]

27 Jul 2025

Dear Dr. Yunus,

Thank you for submitting your manuscript to PLOS ONE. After careful consideration, we feel that it has merit but does not fully meet PLOS ONE’s publication criteria as it currently stands. Therefore, we invite you to submit a revised version of the manuscript that addresses the points raised during the review process.

We look forward to receiving your revised manuscript.

Kind regards,

Li Yang, M.D.

Academic Editor

PLOS ONE

Journal Requirements:

Reviewers' comments:

Reviewer's Responses to Questions

**Comments to the Author**

Reviewer #1: All comments have been addressed

Reviewer #2: All comments have been addressed

Reviewer #5: (No Response)

2. Is the manuscript technically sound, and do the data support the conclusions?

Reviewer #1: Yes

Reviewer #2: Yes

Reviewer #5: Yes

3. Has the statistical analysis been performed appropriately and rigorously?

Reviewer #1: Yes

Reviewer #2: Yes

Reviewer #5: Yes

4. Have the authors made all data underlying the findings in their manuscript fully available?

Reviewer #1: Yes

Reviewer #2: Yes

Reviewer #5: Yes

5. Is the manuscript presented in an intelligible fashion and written in standard English?

Reviewer #1: Yes

Reviewer #2: Yes

Reviewer #5: Yes

Reviewer #1: (No Response)

Reviewer #2: (No Response)

Reviewer #5: The authors partially discuss the clinical significance of the vitamin D/fibrinogen ratio.

As pointed out in the previous section, I believe that the practical significance of utilizing these findings is the core of this paper, so it is necessary to state this clearly.

I believe that adding further detailed explanations will enhance the clinical application value of the research results.

Minor comments

Clinical Value of the Ratio Needs Further Elaboration

The ratio showed a stronger correlation with ACS severity than vitamin D alone, though the difference was not statistically significant. Authors should better clarify the potential clinical utility of this ratio — e.g., how it could complement existing biomarkers like troponin or CRP, and the practicality of dual-assay implementation in triage.

**Do you want your identity to be public for this peer review?** For information about this choice, including consent withdrawal, please see our Privacy Policy

Reviewer #1: No

Reviewer #2: No

Reviewer #5: No

---

## [Author Response · Author response to Decision Letter 3]

14 Aug 2025

We are really thankful to your valuable comments to improve the quality of the manuscript.

Regarding further elaborating the practical significance and clinical application value of the vitamin D/fibrinogen ratio, we have further expanded the clinical implications subsection of the Discussion that while the vitamin D/fibrinogen ratio did not show a statistically significant superiority over vitamin D alone in our cohort, it demonstrated a consistent trend toward stronger correlation with ACS severity and remained an independent predictor after multivariate analysis. We further stated that the ratio integrates two biologically opposing processes, i.e., anti-inflammatory/antithrombotic effects of vitamin D and prothrombotic/pro-inflammatory effects of fibrinogen, into a single index, potentially providing a more holistic measure of thrombo-inflammatory status in ACS.

Minor Comments:

Regarding complementing the established biomarkers, we have added a specific discussion on how the ratio could be incorporated into the current biomarker framework, like troponins or CRP.

Regarding the practicality of Implementation, we have stated that both vitamin D and fibrinogen assays are inexpensive, widely available, and already used in clinical laboratories; hence, in acute triage, this ratio could assist in early identification of higher-risk patients in whom the troponins and CRP are equivocal.

---

## [Decision Letter · Decision Letter 3]

22 Aug 2025

Relationship of Vitamin D, Fibrinogen and Their Ratio with Acute Coronary Syndrome: A Comparative Analysis of Unstable Angina, NSTEMI, and STEMI

PONE-D-25-13346R3

Dear Dr. Yunus,

We’re pleased to inform you that your manuscript has been judged scientifically suitable for publication and will be formally accepted for publication once it meets all outstanding technical requirements.

Kind regards,

Li Yang, M.D.

Academic Editor

PLOS ONE

Additional Editor Comments (optional):

Thanks for the authors' efforts to comprehensively improve your manuscript according to editor's and reviewers' comments. I am pleased to inform you that your paper can be accepted for publication now. Thanks for the chance to assess your work. Additionally, many thanks for all the reviewers' precious inputs.

Reviewers' comments:

Reviewer's Responses to Questions

**Comments to the Author**

Reviewer #5: All comments have been addressed

2. Is the manuscript technically sound, and do the data support the conclusions?

Reviewer #5: Yes

3. Has the statistical analysis been performed appropriately and rigorously?

Reviewer #5: Yes

4. Have the authors made all data underlying the findings in their manuscript fully available?

Reviewer #5: Yes

5. Is the manuscript presented in an intelligible fashion and written in standard English?

Reviewer #5: Yes

Reviewer #5: All comments raised in previous rounds have been adequately addressed. The manuscript is now acceptable for publication.

**Do you want your identity to be public for this peer review?** For information about this choice, including consent withdrawal, please see our Privacy Policy

Reviewer #5: No

---

## [Editor Report · Acceptance letter]

PONE-D-25-13346R3

PLOS ONE

Dear Dr. Yunus,

I'm pleased to inform you that your manuscript has been deemed suitable for publication in PLOS ONE. Congratulations! Your manuscript is now being handed over to our production team.

Kind regards,

on behalf of

Dr. Li Yang

Academic Editor

PLOS ONE